# Contamination of Coupling Glass and Performance Evaluation of Protective System in Vacuum Laser Beam Welding

**Yongki Lee [1,2], Jason Cheon [1,*], Byung-Kwon Min [2] and Cheolhee Kim [1,3]** 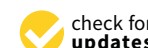

1   Joining Research Group, Korea Institute of Industrial Technology, Incheon 21999, Korea;
    yklee8729@kitech.re.kr (Y.L.); chkim@kitech.re.kr or cheol@pdx.edu (C.K.)
2   Department of Mechanical Engineering, Yonsei University, Seoul 03722, Korea; bkmin@yonsei.ac.kr
3   Department of Mechanical and Materials Engineering, Portland State University, Portland, OR 97201, USA
*   Correspondence: jasonhp@kitech.re.kr; Tel.: +82-32-850-0285

**Abstract:** Vacuum laser beam welding enables deeper penetration depth and welding stability than atmospheric pressure laser welding. However, contaminated coupling glass caused by welding fumes in the vacuum space reduces laser transmittance, leading to inconsistent penetration depth. Therefore, a well-designed protective system is indispensable. Before designing the protective system, the contamination phenomenon was quantified and represented by a contamination index, based on the coupling glass transmittance. The contamination index and penetration depth behavior were determined to be inversely proportional. A cylindrical protective system with a shielding gas supply was proposed and tested. The shielding gas jet provides pressure-driven contaminant suppression and gas momentum-driven contaminant dispersion. The influence of the shielding gas flow rate and gas nozzle diameter on the performance of the protective system was evaluated. When the shielding gas flow was 2.0 L/min or higher, the pressure-driven contaminant suppression dominated for all nozzle diameters. When the shielding gas flow was 1.0 L/min or lower, gas momentum-driven contaminant dispersion was observed. A correlation between the gas nozzle diameter and the contamination index was determined. It was confirmed that contamination can be controlled by selecting the proper gas flow rate and supply nozzle diameter.

**Keywords:** laser welding; vacuum; coupling glass; contamination; shielding gas; protective system; welding consistency

---

## 1. Introduction

Laser beam welding (LBW) employs a high-energy-density power source, and it can achieve relatively deep penetration depths with less heat input than other welding processes [1]. During LBW, deeper penetration depth is enabled by the high energy-per-unit length and/or the high energy density of the laser beam. However, a high energy-per-unit length can lead to keyhole instability and weld defects such as pores, because the high recoil pressure resulting from the high energy can cause melt ejections and frequent keyhole collapses [2]. On the other hand, the small laser beam size required for high energy density can reduce the ability to bridge the welding joint gap, which causes underfill and sagging in the weld cross-section [3].

Many trials have been conducted to achieve both deep penetration and weld quality, and the vacuum laser beam welding (VLBW) method has been proposed as one of the promising methods. VLBW enables deep penetration depth and welding stability under relatively low laser power, which is hard to achieve in atmospheric pressure laser welding [4]. These characteristics are attributed to the

extraordinarily low ambient pressure. As the ambient pressure decreases, the material and welding characteristics that differ from atmospheric pressure are as follows.

Under low ambient pressure, the evaporation temperature of the base material decreases. For steel, for example, the ambient pressure drop from atmospheric pressure to 10 Pa reduces the evaporation temperature from 3100 K to approximately 1850 K. The decrease in vaporization temperature can lower the energy required to vaporize the base metal and conduct keyhole mode welding [5]. In addition, the range between evaporation and melting temperatures (i.e., the temperature range of the liquid phase) decreases because the melting temperature of the metal is almost independent of the ambient pressure. Thin, molten keyhole walls resulting from the narrow temperature range of the liquid phase can enhance keyhole stability by reducing the number of keyhole collapses [6]. Additionally, the size of the plasma plume generated during laser welding decreases, so the energy absorbed into the plume can be reduced. Consequently, keyhole mode welding can be implemented even at low energy input, and a penetration depth more than twice that possible with atmospheric pressure welding can be achieved with VLBW [7–9].

In previous studies, the laser beam was typically delivered through a coupling glass that transmitted the laser beam and sealed the vacuum chamber. However, since VLBW is performed inside a vacuum chamber, the fumes generated during welding adhere to the inner chamber walls and the coupling glass. The contamination of coupling glass caused by the deposition of high-temperature welding fume particles gradually accumulates as the welding trial continues. Since laser energy is absorbed by the contaminated region of the coupling glass, laser transmittance decreases. Furthermore, the contamination and heat generation lead to thermal deformation and changes in the refractive index of the coupling glass, which changes the laser beam profile and the laser focal point [10]. Consequently, penetration depth or weld quality is inconsistent with increasing process time. Therefore, a protective system must be designed and applied to prevent contamination of the coupling glass.

Previous researchers have suggested several protective systems to prevent contamination of the coupling glass in the vacuum chamber. Arata et al. devised two separate vacuum chambers connected vertically, and a single hole on the bottom of the upper chamber acted as an aerodynamic window to prevent contamination of the coupling glass [4]. Katayama et al. applied a cylindrical protective system between the coupling glass and the vacuum chamber. $N_2$ shielding gas was supplied into the cylindrical protective system to prevent contamination of the coupling glass [7]. Jakobs developed a Laser beam welding in Vacuum (LaVa) system to prevent the contamination of coupling glass, where a long, focal-length laser optic and shielding gas were adopted [6].

Until now, most studies on VLBW have focused just on the welding process characteristics. Details of the protective systems and the life cycle of the coupling glass have not been clarified. Therefore, it is currently important to quantitatively measure the contamination of the coupling glass and the influence of factors that determine the performance of the protective system.

In this study, the contamination phenomenon was quantified and visualized using a contamination index based on the transmittance of the coupling glass. Through this analysis, the contamination behavior and its influence on welding quality were verified. A cylindrical protective system was also employed in this study, and the influence of flow rate and nozzle diameter of the shielding gas supply on the performance of the protective system was quantitatively evaluated.

## 2. Materials and Methods

### 2.1. VLBW System Design and Welding Material

Figure 1 shows the laser welding system for VLBW and a magnified schematic of the cylindrical protective system used to address contamination of the coupling glass. The VLBW system is a superordinate concept that includes a cylindrical protective system. The VLBW system is a term used collectively for laser optic, vacuum chamber, and interior construction equipment (coupling glass, positioning stage, and protective system), and a cylindrical protective system that can protect the

coupling glass inside the vacuum chamber. A 1070 nm wavelength laser beam was delivered using a 200 µm diameter fiber and a 300 mm focal length optic. The laser beam was irradiated vertically through a coupling glass into a 74 L capacity vacuum chamber, where the vacuum level was controlled using a rotary pump with a maximum pumping rate of 1500 L/min. During welding, the pressure inside the vacuum chamber was maintained at 10 mbar. The distance between the focal lens of the laser optic and the coupling glass was 145 mm. The laser transmission region between the laser optic and the coupling glass was protected by a bellows, and the beam diameter at the focal point was 356 µm. The diameter of the effective area of the coupling glass was 39 mm, while the laser was transmitted through a 10 mm-diameter area.

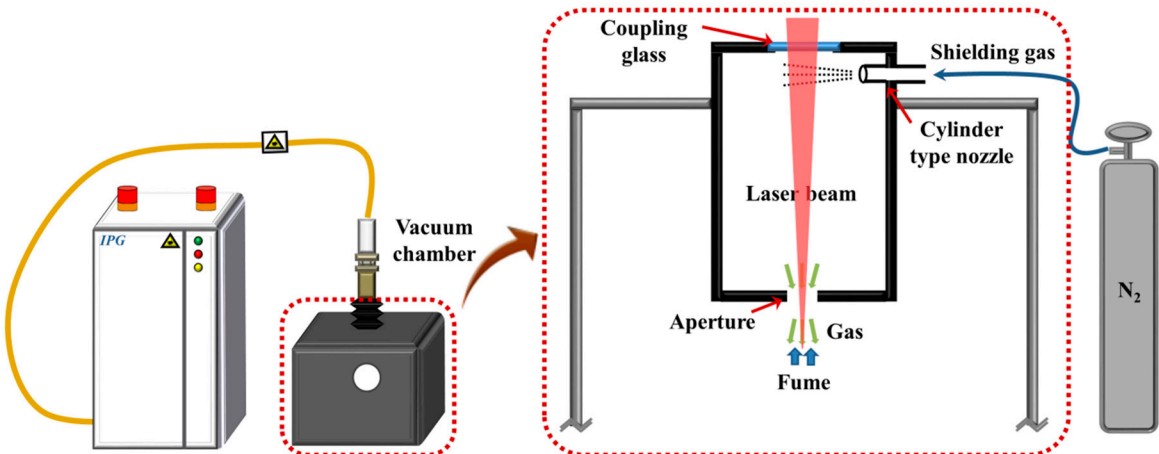

**Figure 1.** Schematic diagram of the vacuum laser beam welding (VLBW) system.

A cylindrical protective system was installed at the bottom of the top surface of the vacuum chamber, as shown in Figure 1. The diameter of the aperture on the bottom of the protective system was chosen to avoid disturbing the laser beam propagation. In this study, $N_2$ shielding gas was supplied by a gas nozzle mounted on the side wall of the protective system.

Figure 2 describes two hypothetical mechanisms of the protective system. The first mechanism is the pressure difference across the aperture. As the shielding gas is continuously supplied from the upper part of the protective system, the internal pressure increases and a local pressure difference is generated across the aperture. If the pressure difference $\Delta P = (P_1 - P_2)$ is positive, the shielding gas flows downward, as shown in Figure 2a, which disturbs the upward motion of the fume particles from the outside to the inside of the vacuum chamber [11]. Where $D_A$ is diameter of the aperture. The second mechanism is the horizontal deflection of the fume particles due to the horizontal shielding gas flow, which is guided by the shielding gas nozzle, as shown in Figure 2b. The shielding gas flow $V_N$ modifies the trajectory of the fume particles out of the coupling glass by changing the velocity of the fume particle $V_F$ into $V_F'$ [12]. Where $D_N$ is diameter of the shielding gas nozzle.

Using the two hypothetical approaches discussed above, contamination of the coupling glass by inflowing fume particles was primarily prevented by the local pressure difference across the aperture. Secondarily, the shielding gas flow was designed to change the trajectory of fume particles toward the outside of the coupling glass, to address fume particles that managed to get into the protective system through the aperture.

The materials used in this experiment were 1.4 mm-thick dual phase (DP) 980 steels with galvanized (GI) coatings. The steel had a tensile strength of 1027 MPa and Zn coating layers on both sides with a weight of 79 g/m$^2$. The specimen size used in this experiment was 70 mm × 70 mm. The vaporization of zinc during welding accelerates fume generation and was used to verify the effect of the process parameters on the contamination of coupling glass.

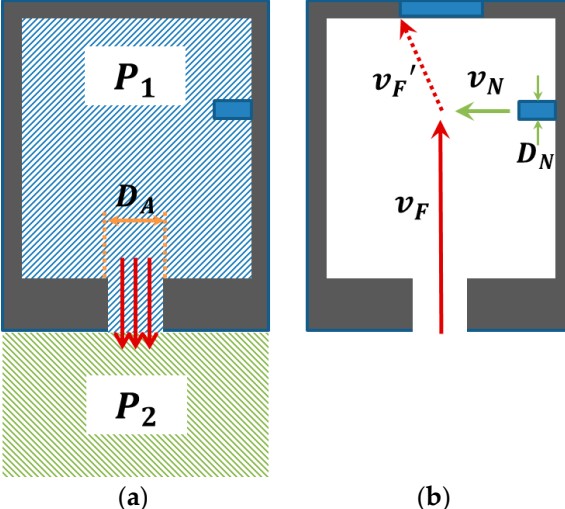

(a)　　　　　　　　(b)

**Figure 2.** The two mechanisms used to prevent fume adhesion to the coupling glass: (**a**) the local pressure difference across the aperture and (**b**) the trajectory change of fume particles.

### 2.2. Experimental Procedure and Analysis Methods

Figure 3 shows the welding path and five sectioning locations used to examine the transverse cross-sections. In order to increase the welding length within the limited chamber space, the welding path was designed to be an Archimedean spiral shape by combining linear translational and rotational motion. The laser beam location can be expressed using the following equation:

$$\begin{aligned} x &= (r_0 - vt)\cos\omega t \\ y &= (r_0 - vt)\sin\omega t \qquad (0\,s \le t \le 90\,s) \end{aligned} \tag{1}$$

where $r_0$ is the x-coordinate of the initial location (60 mm), $v$ is the velocity of the linear translation stage (0.333 mm/s), and $\omega$ is the angular velocity of the rotational stage (0.349 rad/s). Since the welding speed decreases over time, the laser power linearly decreases over time to maintain the energy per unit length at 23.9 J/mm (Figure 4). The total welding length was 1.41 m, and the total welding time was 90 s.

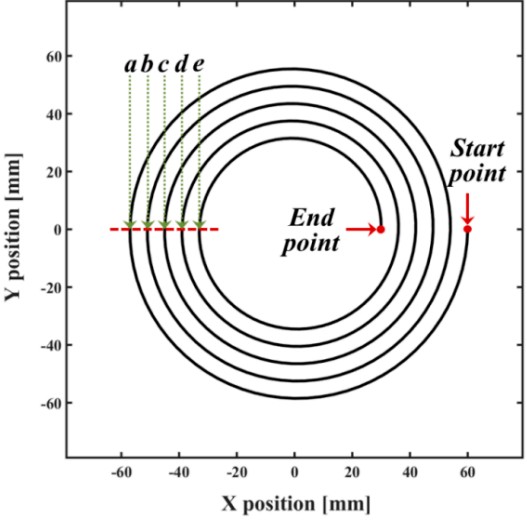

**Figure 3.** Welding shape and sectioning positions (a–e).

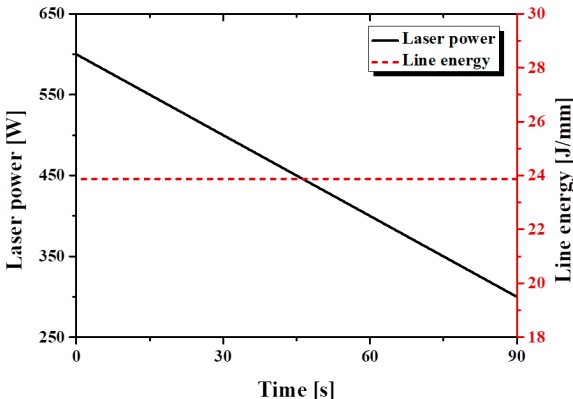

**Figure 4.** Variation in laser power over time for constant line energy.

In the experiments used to analyze the effect of process parameters on the coupling glass contamination and penetration depth, the inner diameter of the cylinder gas nozzle ($D_{nozzle}$) and the shielding gas flow rate were set at three and four levels of control parameters, respectively. Each level of control parameter and the names of the conditions are given in Table 1. The maximum shielding gas flow rate was chosen by considering the maximum pumping rate of the vacuum pump.

**Table 1.** Welding conditions.

| Name | $D_{nozzle}$ [mm] | Flow Rate [L/min] |
|---|---|---|
| D0_F0.0 [1] | - | 0.0 |
| D4_F0.5 | | 0.5 |
| D4_F1.0 | 4 | 1.0 |
| D4_F2.0 | | 2.0 |
| D4_F4.0 | | 4.0 |
| D6_F0.5 | | 0.5 |
| D6_F1.0 | 6 | 1.0 |
| D6_F2.0 | | 2.0 |
| D6_F4.0 | | 4.0 |
| D8_F0.5 | | 0.5 |
| D8_F1.0 | 8 | 1.0 |
| D8_F2.0 | | 2.0 |
| D8_F4.0 | | 4.0 |

[1] Reference case.

In order to visualize and quantify the contamination during the VLBW process, the transmittance of the coupling glass was measured using a spectrum transmission meter, which has an IR peak wavelength of 940 nm and a resolution of 0.1%. The wavelength selected was the highest wavelength available in the spectrum transmission meter and the wavelength nearest to the laser wavelength (1070 nm). Transmittance was measured within the process laser transmission region, the diameter of which was 10 mm (Figure 5a). A total of 41 transmittance values were recorded at 1 mm intervals in eight radial directions from the center, as shown in Figure 5b. The measured transmittance was converted into a contamination index, as follows:

$$\text{Contamination Index [\%]} = 1 - \text{Transmittance [\%]} \qquad (2)$$

Figure 5c indicates the analysis domains that were divided to define the spatial distribution of contamination. If the welding fume rises vertically, the center of the coupling glass will be the most contaminated, and the degree of contamination will be lower toward the periphery. Therefore, it was expected that the contamination of the coupling glass by the welding fume would follow the

standard normal distribution curve. Accordingly, they were divided into three domains, such as sigma zone. A domain was set every $1/3\ r_i$, where $r_i$ is the radius of the process laser transmission region, 5 mm. The median value of the contamination indexes in each domain was used to represent contamination within the domain. To analyze the correlation between the contamination index and the welding consistency, the penetration depths of the welding specimens were measured after the 1st welding trial (W1) and the 10th welding trial (W10), respectively. The sampling locations for five cross-sections are indicated as the positions (a) to (e) in Figure 3. In addition, the relative penetration depth ($S_r$ [%] = (S10/S1) × 100), where S1 and S10 are the penetration depths on the welding samples for W1 and W10, respectively, was calculated to evaluate the welding consistency to evaluate the process parameters of the flow rate and the $D_{nozzle}$.

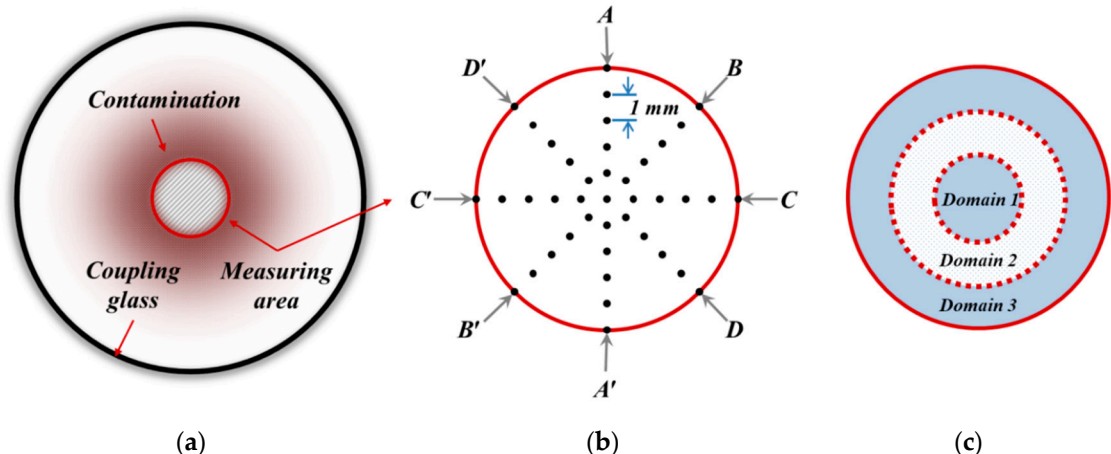

**Figure 5.** Contamination analysis of the coupling glass: (**a**) measuring region, (**b**) transmittance measuring points, and (**c**) analysis domains.

## 3. Results and Discussion

### 3.1. Contamination Behavior

#### 3.1.1. Distribution of Contamination on the Coupling Glass

Figure 6 shows the contaminated coupling glass and contamination index distribution after W10 under the reference parameter set (D0_F0.0) in Table 1. After welding, the laser transmission region on the coupling glass turned an opaque gray color, and its surrounding area became a brown color (Figure 6a). The contamination index for all measured points ranged from 13.4% to 42.6%. Domain 1 showed the highest contamination with 41.2% of the median contamination index. Domain 2 showed 39.5%, and domain 3 showed 28.7% of the median contamination index (Figure 6b).

During the welding trial, welding fumes adhered to the coupling glass, and some fume particles with weak adhesion detached from the coupling glass during the break between the welding trials. For the reference condition, shielding gas was not supplied, and the distribution of contamination on the coupling glass was similar to the velocity distribution of fume particles normalized to the center, as shown in Figure 6b. The contamination accumulated over the welding trial due to the successive adhesion of fume particles. As this occurred, the contamination on the coupling glass absorbed laser energy and raised the temperature of the adhered particles, which accelerated the contamination by further enhancing adhesion.

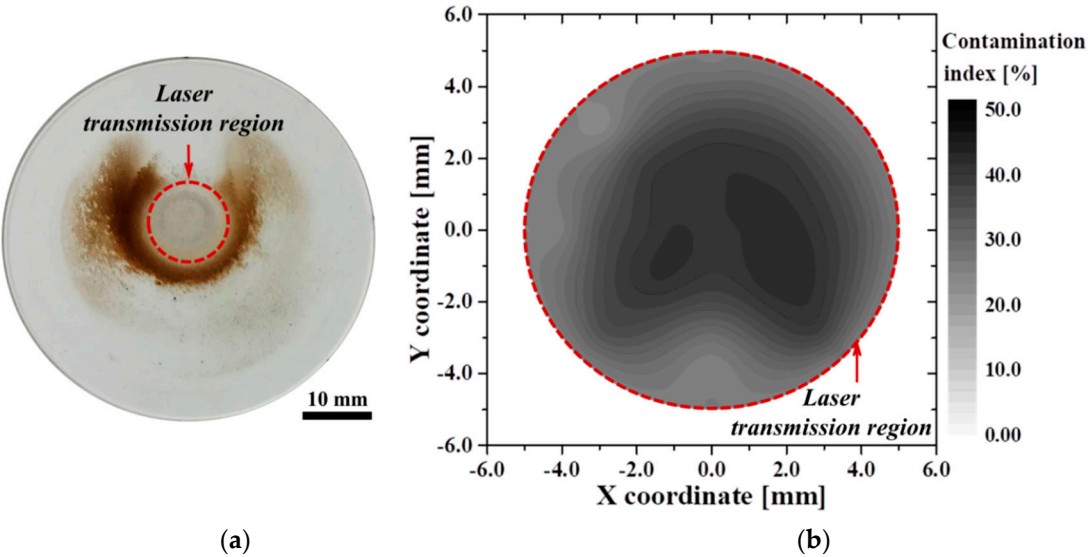

**Figure 6.** The contamination of the coupling glass after the 10th welding trial under the reference condition: (**a**) image of the coupling glass and (**b**) contamination index distribution within the laser transmission region.

3.1.2. Penetration Depth Variation According to Location and Repetitive Welding Trials

For the reference parameter set, when shielding gas was not supplied, obvious differences in penetration depth could be observed along the location, as shown in the cross-sections of the W1 sample (Figure 7a). Although a new coupling glass without contamination was installed before the W1 trial, the glass became contaminated over the welding time, and the penetration depth continuously decreased. At the last measuring point, the measured penetration depth was 0.75 mm, which is only 68.8% of the maximum penetration depth of 1.09 mm.

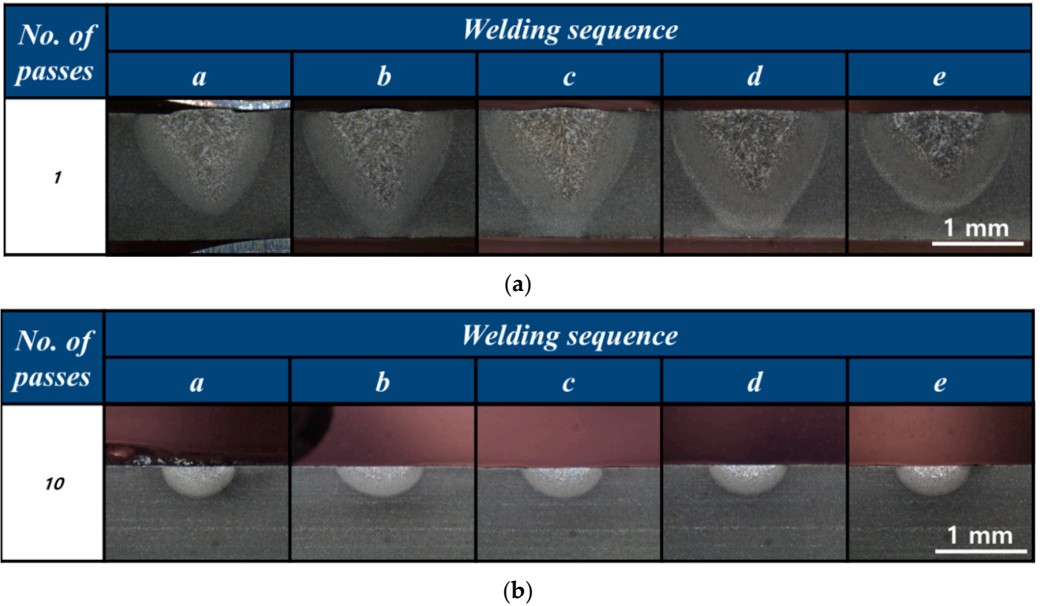

**Figure 7.** Penetration depth according to location: (**a**) 1st welding trial (W1) samples and (**b**) 10th welding trial (W10) samples.

The contamination accumulated continuously with the weld trials, and a penetration depth of 0.17 mm was obtained for the W10 trial, which was only 15.6% of the penetration depth observed for the W1 sample (Figure 7b). The results showed a severe reduction in penetration depth during the

welding trial and between welding trials, which caused inconsistent welding quality that would not be acceptable in most applications. This confirmed that the contamination of the coupling glass needs to be controlled to maintain uniform weldability. In addition, the penetration depth from cross-section (a) to cross-section (e) of W10 is almost the same, indicating that the contamination of the coupling window does not increase indefinitely with time but seems to saturate to a certain value.

### 3.2. Evaluation of Process Parameters

3.2.1. Effect on Contamination of the Coupling Glass

Figure 8 shows the contamination index distributions in the laser transmission region after W10, under all experimental conditions except the reference case. The contamination index in each domain was quantified and is presented in Figure 9. The symbols denote the median for each domain, and the error bar denotes the first and third quartile of the measured data.

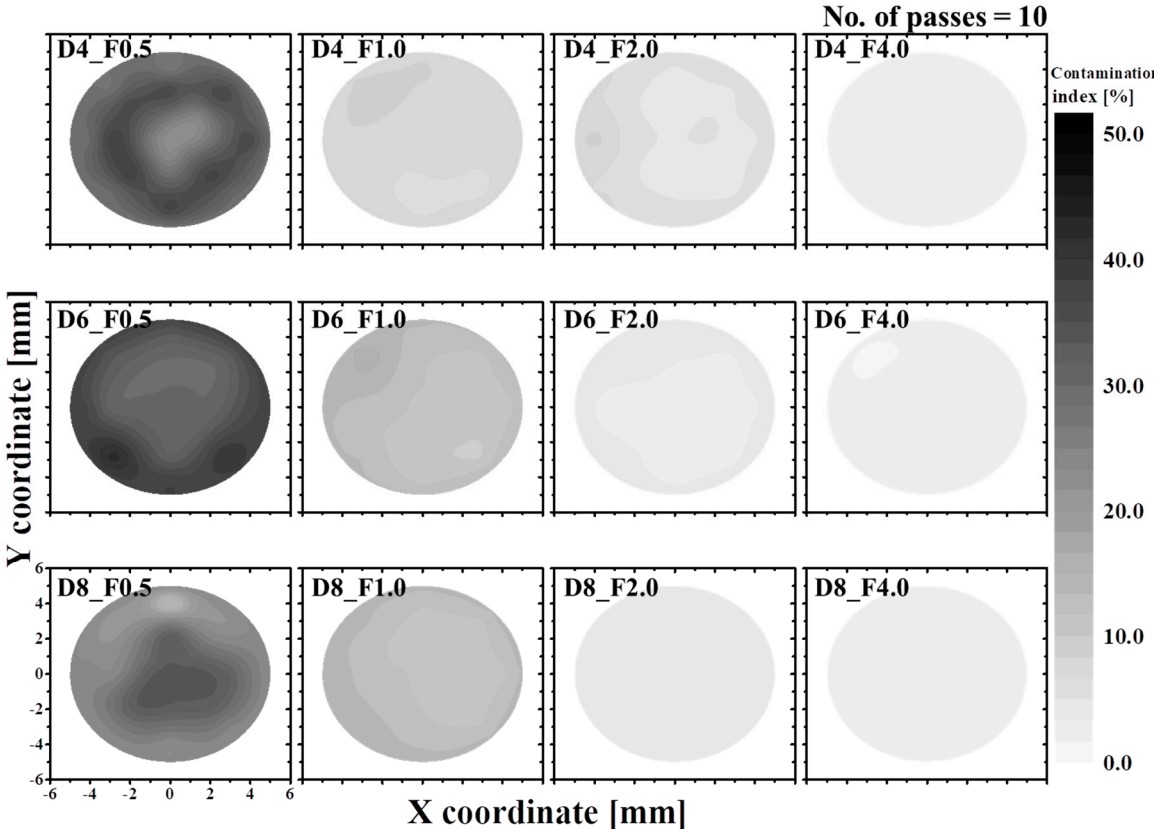

**Figure 8.** Contamination index distribution according to the cylinder gas nozzle ($D_{nozzle}$) and shielding gas flow rate.

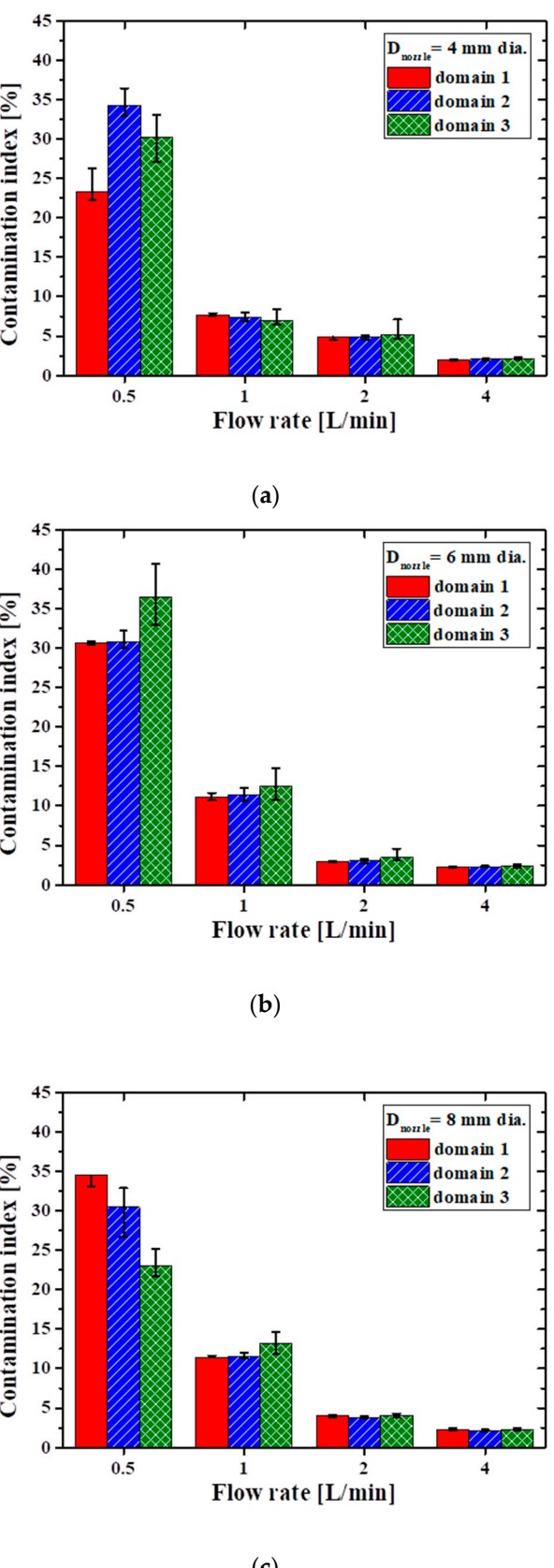

**Figure 9.** Contamination according to $D_{nozzle}$ and shielding gas flow rate: (**a**) $D_{nozzle}$ of 4 mm diameter, (**b**) $D_{nozzle}$ of 6 mm diameter, and (**c**) $D_{nozzle}$ of 8 mm diameter.

As shown qualitatively in Figure 8, the higher the flow rate, the smaller the contamination index. By increasing the shielding gas flow rate, the pressure difference across the aperture of the protective system increases, and the adhesion of particles intruding into the protective system decreases, which is the first protective mechanism explained in Section 2.1. When the flow rate was 2.0 L/min or more, a lower-than-5.2% contamination index was exhibited for all domains, and the difference in contamination index according to the change in $D_{nozzle}$ was less than 2.0%.

When the flow rate was 0.5 L/min, which was the lowest rate in the experiment, the distribution of contamination was influenced by $D_{nozzle}$ because the horizontal shielding gas flow affects the trajectory of the fume particles inside the chamber. This is the second mechanism explained in Sec. 2.1. When the $D_{nozzle}$ is smaller, the flow rate at the center is higher and the fume trajectory is more deflected. Thus, the median contamination indexes at domain 1 for the D4_F0.5 and D6_F0.5 cases were 23.3% and 30.7%, respectively, while for the reference case it was 41.2%. On the other hand, the contamination indexes of domains 2 and 3 for the D4_F0.5 case had an asymmetric distribution in the direction of the shielding gas flow, rather than a normal distribution. Their medians were 34.3% and 30.2% for domain 2 and domain 3, respectively, which were greater than that of domain 1. For the case D8_F0.5, the contamination index was normally distributed, and the mean contamination index of domain 1 was 34.5%.

As the flow rate increased to 1.0 L/min, the contamination indexes drastically decreased compared with those for the cases with a shielding gas flow of 0.5 L/min. The median contamination indices for D6_F1.0 and D8_F1.0 were approximately 12% for all domains; moreover, the contamination index for D4_F1.0 was 7.3% on average, which confirms that a higher flow rate at the center can reduce the contamination at a flow rate of 1.0 L/min. Additionally, differences between the indices for domains decreased. For example, the mean contamination index for domain 1 for the case D8_D1.0 was only 1.7% lower than that of domain 3.

### 3.2.2. Effect on Relative Penetration Depth

The relative penetration depth $S_r$, defined as the ratio of the penetration depth for the W1 sample to that of the W10 sample, was calculated at the cross-sections (a) and (e), respectively, to evaluate the effect of the shielding gas flow rate and $D_{nozzle}$ (Figure 10).

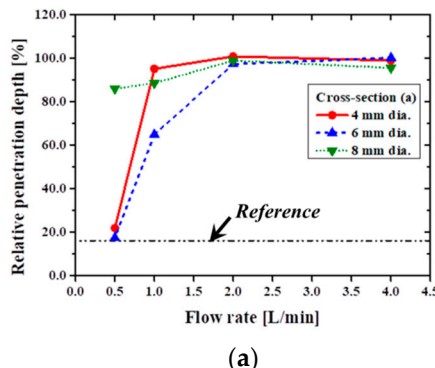
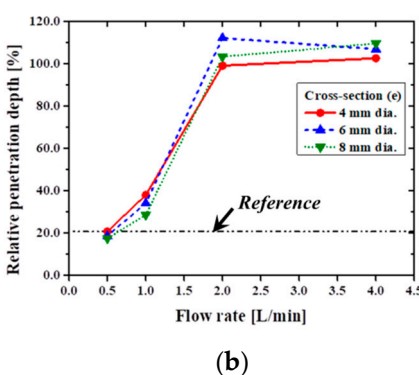

(**a**)    (**b**)

**Figure 10.** Relative penetration depth ($S_r$) according to $D_{nozzle}$ and shielding gas flow rate: (**a**) cross-section (a) and (**b**) cross-section (e).

In contrast to the contamination index, $S_r$ increased with increasing shield gas flow rate. When a shielding gas flow rate of not less than 2.0 L/min was supplied, 95.5% or more $S_r$ could be obtained, as expected from the small contamination indices for those cases shown in Figure 9. When the shielding gas flow rate was 0.5 L/min, the calculated $S_r$ was similar to that of the reference case, and the difference for both was less than 5.81%, except for the cross-section (a), which is an outlier of the D8_F0.5 case. For that case, the contamination index was approximately 30%, and consistent welds were obtained in the welding trials.

When the shielding gas flow was 1.0 L/min, the mean $S_r$ at the cross-section (e) was only 28.9% of that at the cross-section (a). The $S_r$'s for both cross-sections were very similar to the other shielding gas flow rates, except the D8_F0.5 case. Figure 11 shows the relative penetration depths for W10 at all five cross-sections from (a) to (e). The penetration depth tends to decrease from cross-section (a) to cross-section (d) as contamination of the coupling glass during the welding process accumulates. There is an abrupt reduction in penetration depth shown from cross-section (d) to cross-section (e). This originates with the shift in thermal focus with welding time, and it is assumed that a threshold exists between cross-sections (d) and (e) [13].

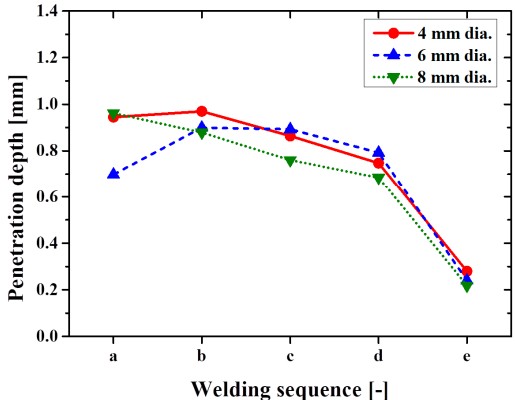

**Figure 11.** The relative penetration depth according to location under a 1.0 L/min shielding gas flow rate.

### 3.2.3. Correlation Between the Contamination Index and the $S_r$

The correlation between the mean contamination index of the coupling glass and $S_r$ is redrawn from the data in Figures 9 and 10, and is shown in Figure 12. For all measured data, the $S_r$ shows an inverse linearity with the contamination index. The Pearson correlation coefficient for both parameters was $-0.99$, which means a strong negative linear correlation. The $S_r$ of 90% or more was maintained when the contamination index of the coupling glass was 7.3% or less.

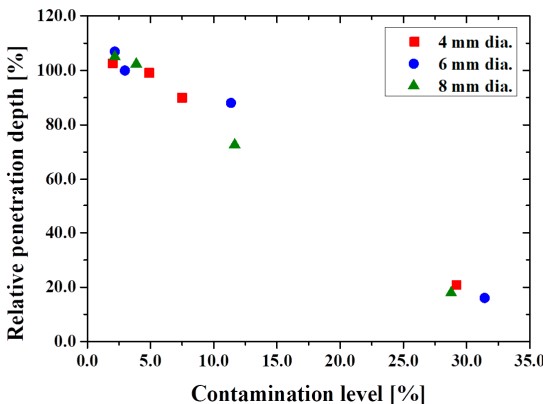

**Figure 12.** Correlation between the contamination index and $S_r$.

In conclusion, we proposed a contamination index that was quantified using a spectrum transmission meter. We experimentally confirmed that it was an effective and accurate index that could be used to represent the contamination of the coupling glass and stability of welding penetration.

## 4. Conclusions

In this paper, the contamination phenomenon was quantified and represented as a contamination index based on transmittance. Through this analysis, the contamination behavior and its effect on

welding quality were analyzed. The influence of the flow rate and the $D_{nozzle}$, which determine the performance of the protective system, were evaluated using the contamination index, and the correlation between the contamination index and welding quality were revealed. As a result, we can present the following conclusions:

- The contamination of the coupling glass was successfully quantified using a contamination index based on measured transmittance. Penetration depth was found to be inversely proportional to the contamination index, which confirms that the contamination index defined in this study can be an effective index.
- Two mechanisms to prevent the contamination of coupling glass were postulated in this study. The first mechanism exploits an applied pressure difference across the aperture, which prohibits the intrusion of fume particles into the protective system. The second mechanism involves the deflection of fume trajectories inside the protective system. Both mechanisms are driven by shielding gas flow.
- When the shielding gas flow was 1.0 L/min or lower, the second mechanism worked in tandem with the first mechanism. An asymmetric pattern of contamination was observed due to the deflection of the welding fume trajectory, and the diameter of the shielding gas nozzle, which determines the flow speed, affected the contamination index.
- When proper shielding gas was not supplied, the coupling glass became contaminated during the welding trial and between welding trials. This caused the penetration depth to vary. The contamination can be controlled by selecting the proper gas flow rate and supply nozzle diameter.

**Author Contributions:** Investigation, Y.L. and J.C.; Methodology, Y.L. and J.C.; Supervision, B.-K.M. and C.K.; Writing—original draft, Y.L. and J.C.; Writing—review and editing, C.K.

**Funding:** This research was funded by the Ministry of Trade, Industry, and Energy, Republic of Korea.

**Conflicts of Interest:** The authors declare no conflict of interest.

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
