# Peer review of "Contamination of Coupling Glass and Performance Evaluation of Protective System in Vacuum Laser Beam Welding"

_applsci, doi:10.3390/app9235082_

Round 1

Reviewer 1 Report

This paper quantified and visualized the contamination phenomenon to clarify the details of the protective system and the life cycle of the coupling glass. The paper is well organized and written. A few suggestions are given below:

(1) Line 70-77: In the introduction, three protective systems are introduced. Is that all or are they the typical protective systems?

(2) Line 78: After the illustration of the used several protective systems, the turn of logic is too sudden (most studies …on the welding process characteristics). What did you base to say that many studies focus on the welding process characteristics? Furthermore, why is the VLBW protective system [6] chosen for research here but not the others?

(3) Line 84-85: “A cylindrical protective system was also …”, what is the difference between the cylindrical protective system and VLBW system? The VLBW is investigated here, but why the cylindrical protective system is also used. It would be great to describe more detail in the introduction.

Author Response

Reply to the review report is attached in Word and PDF file.

Reviewer 2 Report

The present manuscript is well organized and informative. I have several minor suggestions as followed.

Introduction

(1) Authors introduced several protective systems applied in VLBW (Line 70-77). Afterwards, authors mentioned that “A cylinder protective system was also employed in this study” (Line 84-85). I suggest authors should provide a more detailed information about protective systems other researchers have been using and what difference have been introduced in the present study if any.

Materials and Methods

(1) They are some type errors. For example, 79g/m2 (line 127) and 70 mm × 70 mm (line 127).

General:

(1) I suggest authors provide some images of the whole system but not the schematics. In addition, "the contamination of the coupling glass" is confusing, and some detailed images will help readers to understand. Overall, the present manuscript is really well written.

Author Response

Thanks for the careful comments.

The reply to the review report is attached in Word and PDF file.

Reviewer 3 Report

Please find the manuscript enclosed with questions, comments and a few suggestions for improving the clarity and depth of discussions of results. 

Author Response

Thanks for the careful comments.

The reply to the review report is attached in PDF file.
